# OMNIBUS DROPOUT FOR IMPROVING THE PROBABILISTIC CLASSIFICATION OUTPUTS OF CONVNETS

## ABSTRACT

While neural network models achieve impressive classification accuracy across different tasks, they can suffer from poor calibration of their probabilistic predictions. A Bayesian perspective has recently suggested that dropout, a regularization strategy popularly used during training, can be employed to obtain better probabilistic predictions at test time (Gal & Ghahramani, 2016a). However, empirical results so far have not been encouraging, particularly with convolutional networks. In this paper, through the lens of ensemble learning, we associate this unsatisfactory performance with the correlation between the models sampled with dropout. Motivated by this, we explore the use of various structured dropout techniques to promote model diversity and improve the quality of probabilistic predictions. We also propose an *omnibus dropout* strategy that combines various structured dropout methods. Using the SVHN, CIFAR-10 and CIFAR-100 datasets, we empirically demonstrate the superior performance of *omnibus dropout* relative to several widely used strong baselines in addition to regular dropout. Lastly, we show the merit of *omnibus dropout* in a Bayesian active learning application.

## 1 INTRODUCTION

Deep neural networks (NNs) achieve state-of-the-art classification accuracy in many applications. However, in real world scenarios like medical diagnosis and autonomous driving, reliable probabilistic predictions are also crucial and need to be considered in assessing performance. Most modern NNs are trained with maximum likelihood to produce point estimates that are often over-confident (Guo et al., 2017). Bayesian techniques can be used with neural networks to obtain well-calibrated predictions (MacKay, 1992; Neal, 2012), yet they suffer from significant computational challenges. Thus, recent efforts have been devoted to making Bayesian neural networks more efficient (Blundell et al., 2015; Chen et al., 2014; Wu et al., 2018). Monte Carlo (MC) dropout (Gal & Ghahramani, 2016a), a cheap approximate inference technique which obtains uncertainty by performing dropout (Srivastava et al., 2014) at test time, is a popular Bayesian method to obtain uncertainty estimates for NNs.

Despite improvements over deterministic NNs, *MC dropout* can still produce over-confident predictions (Lakshminarayanan et al., 2017), particularly with convolutional architectures. In this paper, we propose a simple yet effective solution to this problem. Inspired by the recent success of explicit ensembles of neural networks obtained using random initializations (Beluch et al., 2018), we reiterate the original notion of dropout as "an extreme form of model combination with extensive parameter sharing" (Srivastava et al., 2014), and interpret *MC dropout* as an ensemble of models. Borrowing machinery from ensemble learning, we then attribute the poor performance of *MC dropout* to its limited model diversity compared to that of explicit ensembles. This perspective reveals how structured dropout methods (Ghiasi et al., 2018; Tompson et al., 2015) can improve performance by promoting diversity. While the importance of diversity has been demonstrated by others, prior works consider explicit ensembles of different models. To the best of our knowledge, this is the first paper to examine structured dropout as a way to enhance diversity in an ensemble obtained from a single model. As discussed below, we also propose to combine different structured dropout methods, which we call *omnibus dropout*. We empirically verify that *omnibus dropout* can yield models with superior performance on the SVHN, CIFAR-10 and CIFAR-100 datasets compared to not only *MC dropout*, but also some of the most widely adopted baselines like *deep ensembles* (Lakshminarayanan et al., 2017) and *temperature scaling* (Guo et al., 2017). Furthermore, we demonstrate the merit of better uncertainty estimates in a Bayesian active learning experiment (Gal et al., 2017b).

## 2  RELATED WORK

Dropout was first introduced as a stochastic regularization technique for NNs (Srivastava et al., 2014). Inspired by the success of dropout, numerous variants have been proposed (Wan et al., 2013; Goodfellow et al., 2013; Tompson et al., 2015; Huang et al., 2016; Singh et al., 2016; Gastaldi, 2017; Ghiasi et al., 2018). Unlike regular dropout, most of these methods drop parts of the NNs in a structured manner. For instance, DropBlock (Ghiasi et al., 2018) applies dropout to small patches of the feature map in convolutional networks, SpatialDrop (Tompson et al., 2015) drops out entire channels, Stochastic Depth Net (Huang et al., 2016) drops out entire ResNet blocks, and Swapout (Singh et al., 2016) combines the Stochastic Depth Net with regular dropout. These methods were proposed to boost test time accuracy. In this paper, we show that these structured dropout techniques can be successfully applied to obtain better uncertainty estimates as well.

Dropout can be thought of as performing approximate Bayesian inference (Gal & Ghahramani, 2016b) and offer estimates of uncertainty. Many other approximate Bayesian inference techniques have also been proposed for NNs (Kingma et al., 2015; Louizos & Welling, 2017). However, these methods can demand a sophisticated implementation, are often harder to scale, and can suffer from sub-optimal performance (Blier & Ollivier, 2018). Another popular alternative to approximate the intractable posterior is Markov Chain Monte Carlo (MCMC) (Neal, 2012). More recently, stochastic gradient versions of MCMC were also proposed to allow scalability (Gong et al., 2019; Ma et al., 2015; Welling & Teh, 2011). Nevertheless, these methods are often computationally expensive, and sensitive to the choice of hyper-parameters. A related approach, the SWA-Gaussian (Maddox et al., 2019) is another technique for Gaussian posterior approximation using the Stochastic Weight Averaging (SWA) algorithm (Izmailov et al., 2018).

There are also non-Bayesian techniques to obtain calibrated confidence estimates. For instance, *temperature scaling* (Guo et al., 2017) has been empirically shown to be effective in calibrating the predictions. A related line of work uses an ensemble of several randomly-initialized NNs (Lakshminarayanan et al., 2017). The method, known as *deep ensembles*, requires training and saving multiple NNs. It has also been demonstrated that an ensemble of snapshots of the trained model at different iterations can help obtain better uncertainty estimates (Geifman et al., 2019). Compared to an explicit ensemble, this approach requires training only one model. Nevertheless, models at different iterations must all be saved in order to deploy the algorithm, which can be computationally demanding.

## 3  AN ANALYSIS OF THE PERFORMANCE OF MC DROPOUT

### 3.1  MC DROPOUT AS ENSEMBLES OF DROPOUT MODELS

Let's assume a dataset $\mathcal{D} = (\boldsymbol{X}, \boldsymbol{Y}) = \{(\boldsymbol{x}_i, y_i)\}_{i=1}^{n}$, where each $(\boldsymbol{x}_i, y_i) \in (\mathcal{X} \times \mathcal{Y})$ is *i.i.d.* We consider the problem of k-class classification, and let $\mathcal{X} \subseteq \mathbb{R}^d$ be the input space and $\mathcal{Y} = \{1, \cdots, k\}$ be the label space[1]. We restrict our attention to NN functions $f_{\boldsymbol{w}}(\boldsymbol{x}) : \mathcal{X} \to \mathbb{R}^k$, where $\boldsymbol{w} = \{W_i\}_{i=1}^{L}$ corresponds to the parameters of a network with L-layers, and $W_i$ corresponds to the weight matrix in the i-th layer. We define a likelihood model $p(y|\boldsymbol{x}, \boldsymbol{w}) = \mathrm{softmax}(f_{\boldsymbol{w}}(\boldsymbol{x}))$. Maximum likelihood estimation can be performed to compute point estimates for $\boldsymbol{w}$.

Recently, Gal & Ghahramani (2016a) proposed a novel viewpoint of dropout as approximate Bayesian inference (See Appendix A for a brief review). This perspective offers a simple way to marginalize out model weights at test time to obtain better calibrated predictions, which is called *MC dropout*:

$$p(y = c|\boldsymbol{x}, \mathcal{D}_{\mathrm{train}}) = \int p(y = c|\boldsymbol{x}, \boldsymbol{w}) p(\boldsymbol{w}|\mathcal{D}_{\mathrm{train}}) d\boldsymbol{w} \approx \frac{1}{T} \sum_{t=1}^{T} p(y|\boldsymbol{x}, \boldsymbol{w}^{(t)}), \quad (1)$$

where $\boldsymbol{w}^{(t)} \sim q(\boldsymbol{w}|\mathcal{D}_{\mathrm{train}})$ is assumed to be independently drawn layer-wise weight matrices: $W_i^{(t)} \sim \hat{W}_i \cdot \mathrm{diag}(\mathrm{Bernoulli}(p))$, $\hat{W}_i$ is the parameter matrix learned during training, and $p$ is the dropout rate. In this paper, we view each dropout sample $\boldsymbol{w}^{(t)}$ in Equation 1 corresponding to an individual model in an ensemble, where *MC dropout* is performing (approximate Bayesian) ensemble averaging.

---

[1]Extension to regression tasks is straightforward but left out of this paper.

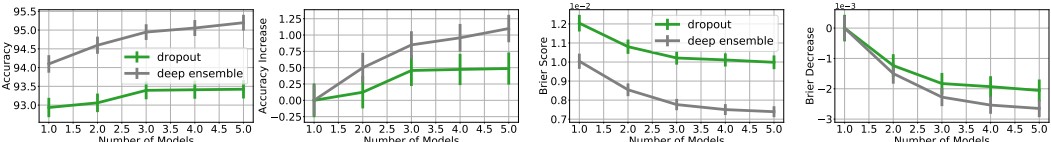

Figure 1: From left to right (1) Accuracy of *MC dropout* and *deep ensemble* (2) the relative improvements in accuracy of *deep ensemble* and *MC dropout* (3) Brier score of *MC dropout* and *deep ensemble* against number of models (4) the relative improvements in in Brier score of *deep ensemble* and *MC dropout* against number of models. Difference grows as number of models increases.

The same analysis applies to structured dropout as well. Mathematically, using structured dropout in lieu of regular dropout amounts to only a change of the approximate distribution $q(\boldsymbol{w}|\mathcal{D}_{\text{train}})$ in Equation 1, so that we are performing Bayesian variational inference with a different class of approximate distributions. For instance, in the channel-level dropout, we sample one Bernoulli random variable for each channel.

## 3.2 DECOMPOSING THE PERFORMANCE OF ENSEMBLES

First proposed by Krogh & Vedelsby (1995), the error-ambiguity decomposition enables one to quantify the performance of ensembles with respect to individual models. Let $\{h_t\}_{t=1}^T$ be an ensemble of $T$ classifiers, and $H(\boldsymbol{x}) = \sum_t h_t(\boldsymbol{x})/T$ is the ensemble prediction. In classification problems, $h_t(\boldsymbol{x})$ is often a probability vector such that $h_t^i(\boldsymbol{x}) = p(y = i|\boldsymbol{x}, \boldsymbol{w}_t)$. In *MC dropout* $h_t(\boldsymbol{x}) = p(y|\boldsymbol{x}, \boldsymbol{w}^{(t)})$. Model ambiguity can be then defined as:

$$\alpha(h_t|\boldsymbol{x}) = ||h_t(\boldsymbol{x}) - H(\boldsymbol{x})||_2^2,$$

which quantifies the difference between an individual model and the ensemble average.

The Brier score measures both the accuracy and calibration of probabilistic classifications, and is proportional to mean squared error (MSE), which can be decomposed for an ensemble as:

$$\text{MSE}(H) = \mathbb{E}_{\boldsymbol{x}}[\text{MSE}(H|\boldsymbol{x})] = \mathbb{E}_{\boldsymbol{x}}[\overline{\text{MSE}}(h|\boldsymbol{x})] - \mathbb{E}_{\boldsymbol{x}}[\overline{\alpha}(h|\boldsymbol{x})], \quad (2)$$

where $\text{MSE}(h_t|\boldsymbol{x}) = ||\boldsymbol{y} - h_t(\boldsymbol{x})||^2$, $\boldsymbol{y}$ is the one-hot encoded vector of the correct label $y$,

$$\overline{\text{MSE}}(h|\boldsymbol{x}) = \frac{1}{T}\sum_t^T \text{MSE}(h_t|\boldsymbol{x}), \text{ and } \overline{\alpha}(h|\boldsymbol{x}) = \frac{1}{T}\sum_t^T \alpha(h_t|\boldsymbol{x})$$

correspond to the average MSE, and ensemble diversity (average ambiguity), respectively. Equation 2 suggests that the more accurate and the more diverse the models, the better performance will be achieved by the ensemble. We use MSE instead of the negative log likelihood (NLL), another commonly used measure for quality of uncertainty estimates, due to mathematical convenience. The two metrics are closely related, and insights obtained from MSE carry over to NLL. In general, MSE or NLL can be seen as comprehensive measures influenced by both the accuracy and the calibration of the model. We give a brief discussion in Appendix B on the relationship between these metrics.

## 3.3 PERFORMANCE OF MC DROPOUT AND MODEL DIVERSITY

The discussion of the previous section provides us with a potential recipe to enhance *MC dropout*. To illustrate the importance of diversity, we conduct an experiment using ResNet-50 on CIFAR-10 to compare *MC dropout* with an explicit ensemble of five NNs (details can be found in Section 4). As we see from Figure 1, individual models in *deep ensemble*, on average, perform better than the ones in *MC dropout*, likely because of the reduced effective capacity of the latter. Furthermore, the performance of the ensembles improve with more models. Yet the improvement is larger for *deep ensemble*, mainly because of increased ensemble diversity, since we know from Equation 2 that the decrease in the Brier score in this analysis is attributable to the increase in ensemble diversity, as the average MSE does not change with the number of models.

The lack of diversity among *MC dropout* models is largely because neighboring pixel features are often correlated in convolutional layers. Thus, even with dropout, similar information propagates

through the network in every iteration. Although we can encourage model diversity naively by increasing dropout rates, this can lead to reduced MSE of individual models, thereby hampering ensemble performance, as can be seen from Eq. 2. This is because higher dropout rates would lead to smaller effective model capacities given a fixed total number of model parameters.

### 3.4 OMNIBUS DROPOUT

While model diversity can be promoted via explicit ensembles, they demand much more computational resources during training, which can be prohibitively expensive. Though typically more number of samples is needed for dropout based methods at test time, unlike *deep ensembles*, dropout uncertainty can be obtained sequentially, which has a lower memory requirement. Moreover, the number of samples needed can potentially be optimized with an adaptive sampling scheme (Inoue, 2019).

In order to enhance diversity in an ensemble obtained from a single model, we examine the use of structured dropout, which drops information from contiguous regions of feature maps so that more divergent information is propagated to subsequent layers during training at each iteration. This enhancement in diversity of predictions can in turn lead to better performance. Specifically, we compare dropout at the *patch-level* which randomly drops out small patches of feature maps (Ghiasi et al., 2018), the *channel-level* which drops out entire channels of feature maps at random (Tompson et al., 2015), and *layer-level* which drops out entire layers of CNNs at random (Huang et al., 2016). We denote these as *dropBlock*, *dropChannel* and *dropLayer* respectively. We identify the test-time sampling of models trained with the aforementioned structured dropout methods as *MC dropBlock*, *MC dropChannel*, *MC dropLayer*.

As we empirically observe below, similar to increasing the dropout rate, the increased diversity of structured dropouts can come at the cost of reduced performance of individual models. Moreover, given considerable choices of dropout strategies available, it can be hard to pick the best one. Therefore, we propose a novel *omnibus dropout* strategy, which merely combines all the aforementioned methods. The implementation of *omnibus dropout* involves the sequential execution of the nested group of dropout methods: regular dropLayer, dropChannel, dropBlock and regular dropout. In our experiments, we use a constant dropout rate for all the dropout methods. Empirically we find this simple choice to mostly work well. As our results show, *omnibus dropout* yields good performance by promoting model diversity without hampering the performance of individual models.

## 4 EXPERIMENTS

We empirically evaluate the performance of *MC dropBlock*, *MC dropChannel*, *MC dropLayer* and *MC omnibus-dropout*, and compare them to *MC dropout*, *deep ensembles* and *temperature scaling*[2]. Unless otherwise stated, the following experimental setup applies to all of our experiments.

**Model.** Layer-level dropout requires skip connections so that there is still information flow through the network after dropping out an entire layer. Some of the examples include the FractalNet (Larsson et al., 2017) and the ResNet (He et al., 2016a). We use the PreAct-Resnet (He et al., 2016b) for all our experiments. We refer to the preAct-ResNet trained without dropout as a *deterministic* model. *MC dropout*, *MC dropBlock* and *MC dropChannel* models are implemented through inserting the corresponding dropout layers with a constant $p$ before each convolutional layer. A block size of $3 \times 3$ is used for *MC dropBlock*. We follow Ghiasi et al. (2018) to match up the effective dropout rate of *MC dropBlock* to the desired dropout rate $p$. *MC dropLayer* is implemented through randomly dropping out entire ResNet blocks at a constant rate $p$. We empirically observe that, dropping out downsampling ResNet blocks during testing is harmful to the quality of uncertainty estimates. This is in agreement with experiments of Veit et al. (2016)[3]. Hence, downsampling blocks are only dropped out during training. *MC omnibus-dropout* is implemented by including all types of aforementioned dropouts, each with the same dropout rate. For a full Bayesian treatment, we also insert a dropout layer before the fully connected layer at the end of the NNs. For all our experiments, the dropout rate of this layer is set to be $0.1$. To ensure a fair comparison, this layer was included for the "deterministic" models. For all models with dropout of all types, we sample 30 times at test-time for Monte Carlo estimation. We implement *deep ensembles* by training five NNs with random initializations. Although

---

[2]See Appendix C for results on explicit dropout ensembles.

[3]In their experiments, ResNet blocks are only dropped out during testing, but not training.

Figure 2: Interrater Agreement (IA) of models with different types of dropout with $0.1$ dropout rate on the SVHN, CIFAR-10 and -100 datasets. The lower the IA, the more diverse the predictions of the models. Y-axis indicates different methods. *MC dropout* produces models with much larger IA, hence less model diversity, than structured dropout techniques in most of the cases.

used for training *deep ensembles* in the original paper, we find that adversarial training hampers both calibration and classification performance significantly, and thus do not incorporate it in our training.

**Datasets.** We conduct experiments using the SVHN (Netzer et al., 2011), CIFAR-10 and CIFAR-100 (Krizhevsky, 2009) datasets with standard train/test-set split. Validation sets of 10000 and 5000 samples are used for SVHN and the CIFARs. To examine the performance of the proposed methods with models of different depth, we use the 18-, 50- and 101-layer PreAct-ResNet for SVHN, CIFAR-10 and CIFAR-100.

**Training.** We perform preprocessing and data augmentation using per-pixel mean subtraction, horizontal random flip and $32 \times 32$ random crops after padding with 4 pixels on each side. We used stochastic gradient descent (SGD) with $0.9$ momentum, a weight decay of $10^{-4}$ and learning rate of $0.01$, and divided it by 10 after 125 and 190 epochs (250 in total) for SVHN and CIFAR-10, and after 250 and 375 (500 in total) for CIFAR-100.

**Evaluation.** All the results are computed on the test set using the model at the optimal epoch based on validation accuracy. We use the Brier score, negative log-likelihood (NLL), expected calibration error (ECE), and Classification accuracy to evaluate performance (see Appendix B for definitions). Following Naeini et al. (2015), we partition predictions into 20 equally spaced bins and take a weighted average of the bins' accuracy and confidence difference to estimate ECE. To visualize calibration performance, we also plot the reliability diagrams (Maddox et al., 2019), which are plots of the difference between accuracy and confidence against confidence. The closer the curve to the X-axis, the more calibrated the model predictions are.

### 4.1 Ensemble Diversity

We first investigate model diversity achieved with dropout. For a fair comparison, we fix the dropout rate for all methods to $0.1$ so that all models have the same effective number of parameters. There are numerous measures that quantify diversity of model ensembles (Zhou, 2012). We use Interrater Agreement (IA) Kuncheva & Whitaker (2003), defined as:

$$\kappa = 1 - \frac{\frac{1}{T} \sum_{k=1}^{n} \rho(x_k)(T - \rho(x_k))}{n(T-1)\bar{p}(1-\bar{p})}, \tag{3}$$

where $T$ is the number of individual classifiers, $n$ is the number of test samples, $\rho(x_k)$ is the number of models that classify the $k$-th sample correctly, and $\bar{p}$ is average classification accuracy across classifiers. When all classifiers perfectly agree on the test set $\kappa = 1$, and smaller values indicate more diverse predictions. Figure 2 summarizes IA for sampled models trained on different datasets with different dropout methods. We also compare the results with *deep ensemble*. The number of models used to compute IA, $T$, is fixed to five for all approaches. In general, IA for *MC dropout* is much higher than structured dropout techniques. On the other hand, structured dropout can yield ensembles that are as diverse as the computationally expensive method of *deep ensemble*, confirming our expectation that dropping out correlated information can produce sampled models with more ambiguity. Note that the large IA for *MC dropLayer* on SVHN is likely caused by a relatively small model used for that problem - an 18-layer ResNet. Lastly, note that while *MC omnibus-dropout* yields models much more diverse than *MC dropout*, it is often not the most diverse one either.

The moderate diversity of *MC omnibus-dropout*, we believe, is the key to its effectiveness. To better understand its behavior, we study the performance metrics as a function of number of sampled models

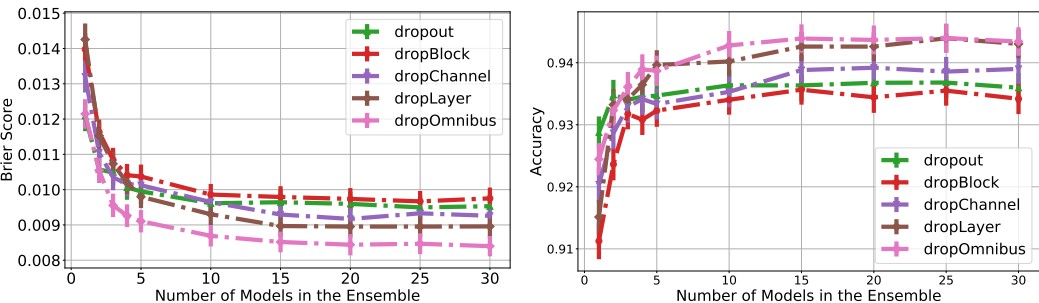

Figure 3: Test Brier score (left) and accuracy (right) against number of models for ensemble prediction at test time on CIFAR-10. This corresponds to the number of different MC dropout instantiations at test time of the same model. The Model trained with *omnibus dropout* achieves the best in terms of accuracy and Brier score.

in the ensemble. Figure 3 shows the Brier score (left) and accuracy (right) against number of models for the CIFAR-10 dataset (Similar results observed for SVHN and CIFAR-100. See Appendix C). Firstly, as seen from Figure 3 (left), while the performance of individual models sampled from *MC dropout* is one of the best, the gain in Brier score with a larger number of test-time MC samples is much smaller compared to structured dropout techniques. On the other hand, though a larger diversity indeed leads to much sharper improvements as number of sampled models increases, the Brier scores (hence MSE) of individual models sampled from *MC dropBlock*, *MC dropChannel* and *MC dropLayer* are much larger than that of *MC dropout*, suggesting a trade-off between diversity and the performance of individual sample models. *MC omnibus-dropout* which enjoys the benefits from both structured and regular dropouts, is able to not only achieve good performance on one sampled model (with a Brier score close to *MC dropout*), but also good model diversity as evident by a significantly larger decrease in Brier score as number of models increases. Similar observations can be made from the accuracy plot of Figure 3 (right).

## 4.2 PERFORMANCE EVALUATION

Table 1 summarizes the performance metrics produced by various models. To ensure a fair comparison, we treat the dropout rate as a hyper-parameter and conduct a linear grid search with $0.05$ interval for optimal dropout rate based on NLL. The optimal dropout rates are shown in the table next to methods. Standard deviations are obtained on five models with random initializations for all dropout models, and bootstrapping the test sets for *deep ensembles*. As seen from Table 1 and Figure 4, all forms of structured dropout models offer better uncertainty estimates than *MC dropout* in general. Overall, *MC omnibus-dropout* and *deep ensembles* are the best performing models. Remarkably, *MC omnibus-dropout* achieves even better uncertainty estimates on SVHN and CIFAR-10 and very comparable ones on CIFAR-100 compared to *deep ensembles* which requires five times more computational resource to train. Moreover, we also perform experiments with five explicit ensembles of models trained together with all types of dropout for a fair comparison, and most of the dropout models outperform *deep ensembles* trianed without dropout. Again, *omnibus dropout* is consistently one of the best methods (See Appendix C). Lastly, as evident from moderately increased classification accuracy over deterministic temperature scaling models, all types of dropout methods can be incorporated into architectures for uncertainty estimates with no accuracy penalty.

We believe the relatively good performance of *MC dropout* on SVHN compared to CIFARs is because the former task is easier so that the model can still predict accurately at an aggressive dropout rate of $0.35$ at which even regular dropout can produce acceptably diverse sampled models. In contrast, as observed in our experiments, while using larger dropout rates for the more difficult CIFAR datasets can lead to more calibrated predictions, accuracy and NLL suffer due to drop in MSE of individual models (see Appendix C). Lastly, we believe the results for *MC dropBlock* can be improved by optimizing the choice of block size. A pre-fixed block size of $3 \times 3$ can be too small for the upstream convolutional layers where the size of feature maps are much larger than the block size, and too large for the last few downstream layers where the feature maps are comparable to the block size, as supported by sharp increases in NLL after the optimal dropout rate.

Table 1: Results on benchmark datasets comparing accuracy and uncertainty estimates produced by different types of methods. The top-2 performing results for each metric are bold-faced. *MC omnibus-dropout* is consistently one of the best methods, outperforming even *deep ensembles*, which requires five times more computational resources, in many cases. The numbers in bracket next to dropout methods corresponds to the optimal $drop\_rate$ found by grid search using the NLL metric.

| Datasets | Methods | Accuracy ↑ | NLL ↓ | Brier ↓ $(\times 10^{-3})$ | ECE ↓ $(\times 10^{-2})$ |
|---|---|---|---|---|---|
| SVHN | Temp Scaling | $95.7 \pm 0.1$ | $0.163 \pm 0.002$ | $6.62 \pm 0.10$ | $0.995 \pm 0.160$ |
| | Deep Ensemble | $96.6 \pm 0.1$ | $0.179 \pm 0.009$ | $5.39 \pm 0.16$ | $1.08 \pm 0.08$ |
| | Dropout (0.35) | $96.7 \pm 0.1$ | $\mathbf{0.128 \pm 0.001}$ | $\mathbf{5.11 \pm 0.06}$ | $0.934 \pm 0.045$ |
| | DropBlock (0.1) | $\mathbf{96.8 \pm 0.1}$ | $0.133 \pm 0.002$ | $5.19 \pm 0.07$ | $1.26 \pm 0.14$ |
| | DropChannel (0.2) | $96.7 \pm 0.1$ | $0.130 \pm 0.001$ | $5.15 \pm 0.06$ | $\mathbf{0.799 \pm 0.032}$ |
| | DropLayer (0.25) | $96.3 \pm 0.1$ | $0.144 \pm 0.002$ | $5.69 \pm 0.05$ | $\mathbf{0.846 \pm 0.250}$ |
| | Omnibus dropout (0.15) | $\mathbf{96.9 \pm 0.1}$ | $\mathbf{0.127 \pm 0.001}$ | $\mathbf{4.97 \pm 0.09}$ | $1.15 \pm 0.06$ |
| CIFAR10 | Temp Scaling | $93.9 \pm 0.1$ | $0.189 \pm 0.002$ | $9.06 \pm 0.08$ | $0.905 \pm 0.114$ |
| | Deep Ensemble | $\mathbf{95.2 \pm 0.2}$ | $\mathbf{0.181 \pm 0.009}$ | $\mathbf{7.40 \pm 0.28}$ | $1.40 \pm 0.16$ |
| | Dropout (0.2) | $93.1 \pm 0.1$ | $0.224 \pm 0.003$ | $10.2 \pm 0.1$ | $1.64 \pm 0.07$ |
| | DropBlock (0.1) | $93.4 \pm 0.1$ | $0.203 \pm 0.003$ | $9.89 \pm 0.10$ | $\mathbf{0.743 \pm 0.116}$ |
| | DropChannel (0.15) | $93.7 \pm 0.1$ | $0.193 \pm 0.002$ | $9.34 \pm 0.9$ | $0.812 \pm 0.104$ |
| | DropLayer (0.1) | $94.0 \pm 0.2$ | $0.206 \pm 0.001$ | $9.09 \pm 0.17$ | $0.941 \pm 0.068$ |
| | Omnibus dropout (0.1) | $\mathbf{94.4 \pm 0.1}$ | $\mathbf{0.173 \pm 0.001}$ | $\mathbf{8.38 \pm 0.10}$ | $\mathbf{0.607 \pm 0.078}$ |
| CIFAR100 | Temp Scaling | $74.5 \pm 0.3$ | $1.00 \pm 0.01$ | $3.57 \pm 0.04$ | $4.02 \pm 0.62$ |
| | Deep Ensemble | $\mathbf{77.9 \pm 0.4}$ | $\mathbf{0.922 \pm 0.019}$ | $\mathbf{3.12 \pm 0.05}$ | $5.10 \pm 0.33$ |
| | Dropout (0.2) | $74.1 \pm 0.4$ | $1.18 \pm 0.01$ | $3.71 \pm 0.05$ | $9.18 \pm 0.23$ |
| | DropBlock (0.15) | $73.7 \pm 0.5$ | $1.04 \pm 0.02$ | $3.66 \pm 0.05$ | $4.46 \pm 0.97$ |
| | DropChannel (0.15) | $74.9 \pm 0.5$ | $0.996 \pm 0.02$ | $3.46 \pm 0.04$ | $3.17 \pm 0.11$ |
| | DropLayer (0.25) | $\mathbf{75.7 \pm 0.2}$ | $1.01 \pm 0.01$ | $3.42 \pm 0.03$ | $\mathbf{2.90 \pm 0.24}$ |
| | Omnibus dropout (0.25) | $75.3 \pm 0.2$ | $\mathbf{0.929 \pm 0.005}$ | $3.40 \pm 0.02$ | $\mathbf{1.65 \pm 0.21}$ |

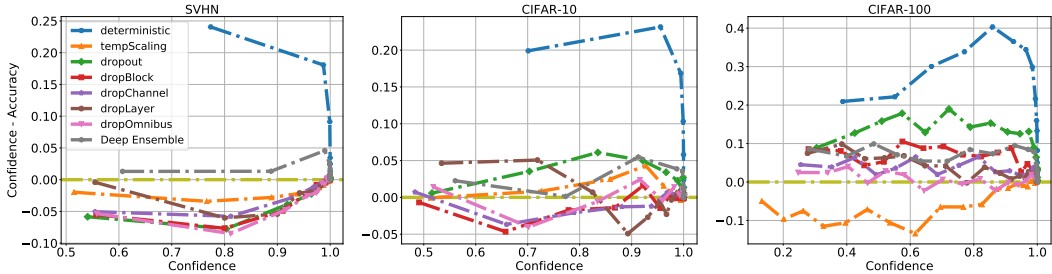

Figure 4: Reliability diagrams of predictions produced by difference models.

## 4.3 Bayesian Active Learning

To further demonstrate the merit of *omnibus dropout*, we consider the downstream task of Bayesian active learning on CIFAR-10. Active learning involves first training on a small amount of labeled data. Then, an acquisition function based on the outputs of models is used to select a small subset of unlabeled data so that an oracle can provide labels for these queried data. Samples that a model is the least confident about are usually selected for labeling, in order to maximize the information gain. The model is then retrained with the additional labeled data that is provided. The above process can be repeated until a desired accuracy is achieved or the labeling resources are exhausted.

In our experiment, we train models with structured dropout at different scales using the identical setup as described in the beginning of this section, except that only 2000 training samples are used initially. To match up model capacity, the dropout rate is set to $0.1$ for all methods. We also compare again a deterministic model. After the first iteration, we acquire 1000 samples from a pool of "unlabeled" data, and combine the acquired samples with the original set of labeled images to retrain the models. Following Gal et al. (2017b), we consider three acquisition functions: *Max Entropy*, $\mathbb{H}[y|\boldsymbol{x}, \mathcal{D}_{train}] = -\sum_c p(y = c|\boldsymbol{x}, \mathcal{D}_{train}) \log p(y = c|\boldsymbol{x}, \mathcal{D}_{train})$, the *BALD* metric (Bayesian Active Learning by Disagreement), $\mathbb{I}[y, \boldsymbol{w}|\boldsymbol{x}, \mathcal{D}_{train}] = \mathbb{H}[y|\boldsymbol{x}, \mathcal{D}_{train}] - \mathbb{E}_{p(\boldsymbol{w}|\mathcal{D}_{train})}[\mathbb{H}[y|\boldsymbol{x}, \boldsymbol{w}]]$, and the *Variation Ratios* metric, variation-ratio$[\boldsymbol{x}] = 1 - \max_y p(y, \boldsymbol{x}, \mathcal{D}_{train})$. We repeat the acquisition

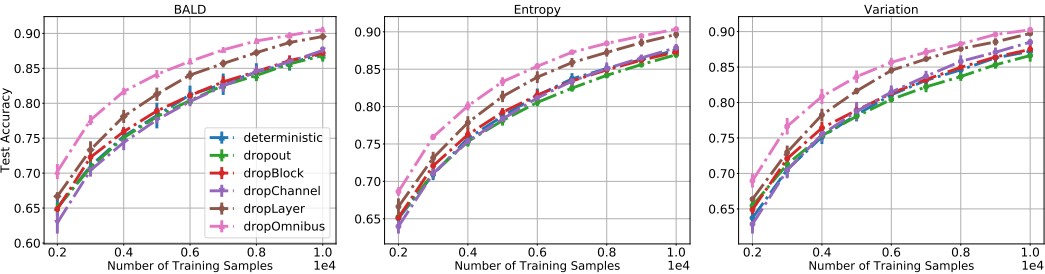

Figure 5: *Left*: Test accuracy against number of training samples for models with different methods of dropout and Variation Ratios as the acquisition function on CIFAR-10. *Right*: Relative improvements in test accuracy over that of the first iteration with different methods of dropout.

process eight times so that in the last iteration, the training set contains 10000 images. To mimic a real world scenario in which number of labeled samples is small, we do not use a validation set, and the accuracies reported for this experiment correspond to the last-epoch accuracies. We repeat experiments five times for consistency.

Figure 5 shows the test accuracy against number of training samples for different models. In general, *MC omnibus-dropout* yields the best performance by far. Interestingly, *MC omnibus-dropout* is able to outperform all other methods consistently by a significant margin after the first iteration when all samples are randomly selected. In addition, it can be seen that, after the first iteration when all 2000 training images are randomly selected, the test accuracy using *MC dropout* is on par with that of other structured dropout methods. However, as more labeled data are added, the relative increase in accuracy is more significant for models using structured dropout compared to that of using regular dropout. This suggests that the uncertainty estimates obtained with structured dropout are more useful for assessing "what the model doesn't know", thereby allowing for the selection of samples to be labeled in a way that better helps improve performance. Note also that the comparative gain in accuracy by *MC omnibus-dropout* during the later stages of the learning process is not as large. We suspect this can be caused by the saturation effect on test accuracy.

## 5 CONCLUSION AND FUTURE WORK

We reinterpret *MC dropout* as ensemble averaging strategy, and attribute its poor performance in convolutional neural networks to a lack of diversity of sampled models using the error-ambiguity decomposition of the Brier score (or MSE), a widely used performance metric that captures both accuracy and calibration of probabilistic outputs. As we demonstrate empirically, *omnibus dropout*, which is simple-to-implement and computationally efficient, strikes the right balance between model diversity among sampled models while retaining reasonable performance of individuals models, thereby consistently improving the quality of the ensemble's prediction.

We are interested in further exploring several directions. Firstly, we have only considered uniform individual dropout rates within the omnibus dropout strategy. Learning the optimal dropout rates for each type of dropout, possibly by building on the the work of (Gal et al., 2017a), can potentially further improve the performance of *omnibus dropout*. Moreover, we used a constant dropout rate in our experiments, even though one can vary dropout rates across NNs (Huang et al., 2016) or incorporate dropout rate scheduling (Ghiasi et al., 2018). How this impacts the quality of the probabilistic predictions is an open question. Lastly, we have only explored structured dropout in the context of CNNs, with application to computer vision tasks. We believe this idea can be extended beyond CNNs.

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

APPENDIX A: BRIEF REVIEW OF DROPOUT AS BAYESIAN APPROXIMATION

Let us assume a dataset $\mathcal{D} = (\boldsymbol{X}, \boldsymbol{Y}) = \{(\boldsymbol{x}_i, y_i)\}_{i=1}^n$, where each $(\boldsymbol{x}_i, y_i) \in (\mathcal{X} \times \mathcal{Y})$ is *i.i.d.* In this paper, we consider the problem of k-class classification, and let $\mathcal{X} \subseteq \mathbb{R}^d$ be the feature space and $\mathcal{Y} = \{1, \cdots, k\}$ be the label space. A classifier is a function that maps input feature space to the label space $f : \mathcal{X} \to \mathbb{R}^c$. We restrict our attention to functions that can be implemented as a DNN, and denote it by $f_{\boldsymbol{w}}(\boldsymbol{x})$, where $\boldsymbol{w} = \{W_i\}_{i=1}^L$ corresponds to the parameters of a network with L-layers, and $W_i$ corresponds to the weight matrix in the i-th layer. We define a likelihood model $p(y|\boldsymbol{x}, \boldsymbol{w}) = \text{softmax}(f_{\boldsymbol{w}}(\boldsymbol{x}))$. It is common practice to perform maximum likelihood to compute point estimates for $\boldsymbol{w}$. Uncertainty estimates can be obtained through Bayesian DNNs by first assuming a prior distribution on the weights, $p(\boldsymbol{w})$. A common choice is the zero mean Gaussian $\mathcal{N}(0, I)$. Bayes Theorem can then be used to obtain the posterior $p(\boldsymbol{w}|\boldsymbol{X}, \boldsymbol{Y}) = p(\boldsymbol{Y}|\boldsymbol{X}, \boldsymbol{w})p(\boldsymbol{X})/p(\boldsymbol{Y}|\boldsymbol{X})$, with which inference can be carried out:

$$p(y = c|\boldsymbol{x}, \mathcal{D}_{\text{train}}) = \int p(y = c|\boldsymbol{x}, \boldsymbol{w})p(\boldsymbol{w}|\mathcal{D}_{\text{train}})d\boldsymbol{w}. \tag{4}$$

The marginal distribution $p(\boldsymbol{Y}|\boldsymbol{X})$, and thus $p(\boldsymbol{w}|\boldsymbol{X}, \boldsymbol{Y})$ are often intractable. Variational inference uses a tractable family of distributions $q_\theta(\boldsymbol{w})$ paramaterized by $\theta$ to approximate the true posterior $p(\boldsymbol{w}|\boldsymbol{X}, \boldsymbol{Y})$, by minimizing the Kullback-Leibler divergence $\text{KL}(q_\theta(\boldsymbol{w})|p(\boldsymbol{w}|\boldsymbol{X}, \boldsymbol{Y}))$, which is equivalent to optimizing a bound on the true objective Graves (2011). To interpret dropout as a variational inference strategy Gal & Ghahramani (2016a), the approximate distribution is defined as:

$$W_i = \Theta_i \cdot \text{diag}(z_{i,j})_{j=1}^{K_i}, \tag{5}$$

$$z_{i,j} \sim \text{Bernoulli}(p_i) \text{ for } i = 1, \cdots, L, \ j = i, \cdots, K_{i-1}, \tag{6}$$

where $\theta = \{\Theta_i\}_{i=1}^L$ are variational parameters to be optimized and $\{p_i\}_{i=1}^L$ are user-defined hyperparameters that correspond to layerwise dropout rates. Minimizing the KL-divergence is mathematically equivalent to maximizing the following objective:

$$\mathcal{L}_{VI}(\theta) = \sum_{i=1}^n \int q_\theta(\boldsymbol{w}) \log p(y_i|\boldsymbol{x}_i, \boldsymbol{w})d\boldsymbol{w} - \text{KL}(q_\theta(\boldsymbol{w})|p(\boldsymbol{w})). \tag{7}$$

Using Monte Carlo integration with one sample $\boldsymbol{w}_i \sim q_\theta(\boldsymbol{w})$ for each training datum $(\boldsymbol{x}, y)$ to approximate the integral in the above equation, and optimizing over mini-batches of size $m$, the approximated objective becomes:

$$\hat{\mathcal{L}}_{VI}(\theta) = \frac{n}{m} \sum_{i=1}^m \log p(y_i|\boldsymbol{x}_i, \boldsymbol{w}_i) - \text{KL}(q_\theta(\boldsymbol{w})|p(\boldsymbol{w})). \tag{8}$$

As shown in Gal & Ghahramani (2016a), there is a direct correspondence between optimizing the above objective and regular dropout training for DNNs. Furthermore, uncertainty estimates can be obtained through marginalizing and performing Monte Carlo integration over the approximate distribution $q_\theta(\boldsymbol{w})$. This corresponds to dropout at test time:

$$p(y = c|\boldsymbol{x}, \mathcal{D}_{\text{train}}) \approx \int p(y = c|\boldsymbol{x}, \boldsymbol{w})q_\theta(\boldsymbol{w})d\boldsymbol{w} \approx \frac{1}{T} \sum_{t=1}^T p(y|\boldsymbol{x}, \boldsymbol{w}_t), \tag{9}$$

where $\boldsymbol{w}_t \sim q_\theta(\boldsymbol{w})$ are dropout samples from the NN. This is referred to as the *MC dropout*.

APPENDIX B: RELATIONSHIP BETWEEN DIFFERENT PERFORMANCE METRICS

Brier score, negative log-likelihood (NLL) and the expected calibration error (ECE) are three of the most commonly used metrics for evaluating the quality of uncertainty estimates. In this section, we discuss the relationship between them.

As we noted in Section 3, the Brier score is equal to the normalized MSE in the context of classification. Recall, the ECE is defined as:

$$\text{ECE}(H) = \mathbb{E}_{\boldsymbol{x}}[(\mathbb{E}_y[y|H(\boldsymbol{x})] - H(\boldsymbol{x}))^2], \tag{10}$$

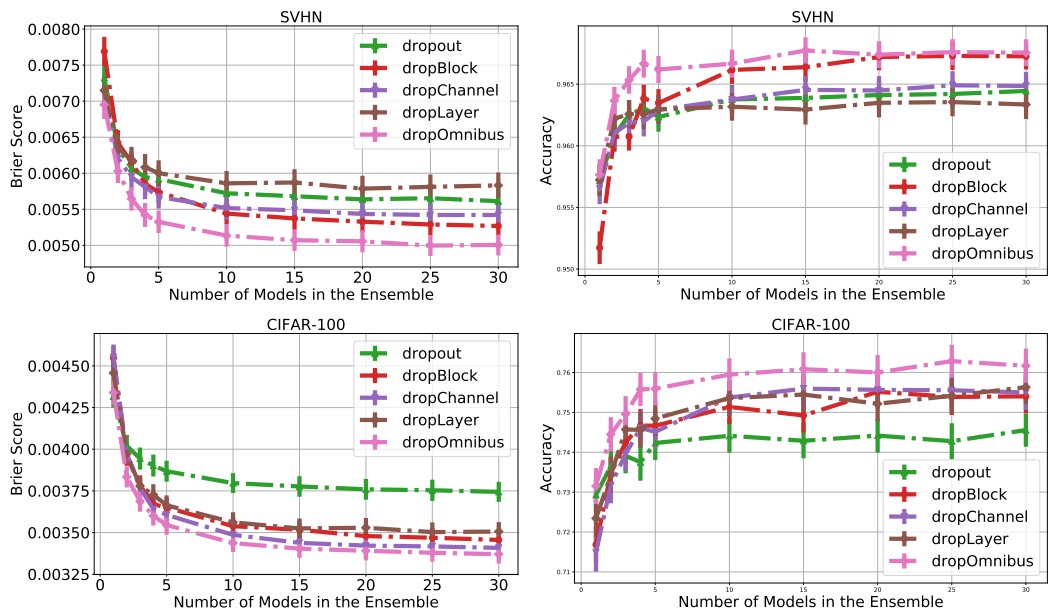

Figure 6: Test Brier score (left) and accuracy (right) against number of models for ensemble prediction at test time on SVHN and CIFAR-100. This corresponds to the number of different MC dropout instantiations at test time of the same model. The Model trained with *omnibus dropout* achieves the best in terms of accuracy and Brier score.

which measures the expected difference between the true class probability and the confidence of the model (Kuleshov & Liang, 2015). In addition to the error-ambiguity decomposition that we have discussed, MSE can also be decomposed as:

$$\text{MSE}(H) = \mathbb{E}_{\boldsymbol{x}}[(y - H(\boldsymbol{x}))^2] \tag{11}$$

$$= \mathbb{E}_{\boldsymbol{x}}[(y - \mathbb{E}_y[y|H(\boldsymbol{x})])^2] + \text{ECE}(H) \tag{12}$$

$$= \text{Var}_{\boldsymbol{x}}[y] - \text{Var}_{\boldsymbol{x}}[\mathbb{E}_y[y|H(\boldsymbol{x})]] + +\text{ECE}(H), \tag{13}$$

where $\mathbb{E}_y[y|H(\boldsymbol{x})]$ corresponds to the true probability of $y = 1$ conditioned on $H(\boldsymbol{x})$. $\text{Var}_{\boldsymbol{x}}[\mathbb{E}_y[y|H(\boldsymbol{x})]]$ measures the variation of the true class probabilities across the level-sets of the ensemble model $H$ Kuleshov & Liang (2015). Thus for this metric, the numeric values of $H(\boldsymbol{x})$ are not important. It is minimized if $H(\boldsymbol{x})$ is a constant and maximized when $H(\boldsymbol{x}) = f(y)$, for any bijective function $f$. One can therefore view $\text{Var}_{\boldsymbol{x}}[\mathbb{E}_y[y|H(\boldsymbol{x})]]$ as a weak metric of accuracy that is not sensitive to calibration. Note $\text{Var}_{\boldsymbol{x}}[y]$ does not depend on the models. Brier score thus can be seen as a metric that is influenced by both the accuracy and the ECE of the models. Similarly, NLL is a metric closely related Brier score on a $\log$ scale. Consequently, sometimes better uncertainty estimates in terms of NLL or Brier score can lead to slight drops in accuracy, as the reduction in calibration error outweighs increase in classification error. This phenomenon is indeed observed in practice as well. Figure 7 shows the plot of both NLL and accuracy against dropout rates for all dropout methods considered in the paper. For instance, it can be seen that while increasing the dropout rate for the *MC dropout* model on CIFAR-100 dataset from $0.1$ to $0.2$ leads to a reduction in NLL, there is also quite a significant dropout in classification accuracy. Similar trends can be seen for *MC dropChannel* on CIFAR-10 as well. Nevertheless, the trade-off is not always present. To exemplify, increasing dropout rate of *MC dropout* on the SVHN dataset also leads to an increase in accuracy as well. In conclusion, when tuning for the optimal dropout rate in practice, it can be beneficial to look at different metrics for a holistic consideration.

## APPENDIX C: ADDITIONAL RESULTS

**Supplementary Results on Diversity of Dropout Models.** In Figure 6, we show plots of Brier score and accuracy against number of models used for prediction on SVHN and CIFAR-100 datasets.

Table 2: Results comparing accuracy and uncertainty estimates obtained using a **single model** when $drop\_rate = 0.1$ for all models. The top-2 performing results for each metric is bold-faced. *MC omnibus-dropout* is the best method in general.

| Datasets | Methods | Accuracy ↑ | NLL ↓ | Brier ↓ ($\times 10^{-3}$) | ECE ↓ ($\times 10^{-2}$) |
|---|---|---|---|---|---|
| SVHN | Temp Scaling | $95.7 \pm 0.1$ | $0.163 \pm 0.002$ | $6.62 \pm 0.10$ | $0.995 \pm 0.160$ |
| | Dropout | $96.4 \pm 0.1$ | $0.179 \pm 0.004$ | $5.68 \pm 0.07$ | $1.34 \pm 0.10$ |
| | DropBlock | $\mathbf{96.8 \pm 0.1}$ | $\mathbf{0.133 \pm 0.002}$ | $\mathbf{5.19 \pm 0.07}$ | $1.26 \pm 0.14$ |
| | DropChannel | $96.5 \pm 0.1$ | $0.148 \pm 0.002$ | $5.41 \pm 0.04$ | $\mathbf{0.663 \pm 0.050}$ |
| | DropLayer | $96.2 \pm 0.1$ | $0.154 \pm 0.002$ | $5.94 \pm 0.10$ | $1.13 \pm 0.10$ |
| | Omnibus dropout | $\mathbf{96.8 \pm 0.1}$ | $\mathbf{0.133 \pm 0.003}$ | $\mathbf{5.07 \pm 0.07}$ | $\mathbf{0.616 \pm 0.077}$ |
| CIFAR10 | Temp Scaling | $93.9 \pm 0.1$ | $\mathbf{0.189 \pm 0.002}$ | $\mathbf{9.06 \pm 0.08}$ | $0.905 \pm 0.114$ |
| | Dropout | $93.8 \pm 0.1$ | $0.226 \pm 0.008$ | $9.44 \pm 0.10$ | $2.30 \pm 0.09$ |
| | DropBlock | $93.4 \pm 0.1$ | $0.203 \pm 0.003$ | $9.89 \pm 0.10$ | $\mathbf{0.743 \pm 0.116}$ |
| | DropChannel | $93.7 \pm 0.1$ | $0.196 \pm 0.006$ | $9.20 \pm 0.136$ | $0.970 \pm 0.171$ |
| | DropLayer | $\mathbf{94.0 \pm 0.2}$ | $0.206 \pm 0.001$ | $9.09 \pm 0.17$ | $0.941 \pm 0.068$ |
| | Omnibus dropout | $\mathbf{94.4 \pm 0.1}$ | $\mathbf{0.173 \pm 0.001}$ | $\mathbf{8.38 \pm 0.10}$ | $\mathbf{0.607 \pm 0.078}$ |
| CIFAR100 | Temp Scaling | $74.5 \pm 0.3$ | $\mathbf{1.00 \pm 0.01}$ | $3.57 \pm 0.04$ | $\mathbf{4.02 \pm 0.62}$ |
| | Dropout | $74.8 \pm 0.4$ | $1.21 \pm 0.01$ | $3.71 \pm 0.05$ | $11.1 \pm 0.4$ |
| | DropBlock | $75.6 \pm 0.2$ | $1.04 \pm 0.01$ | $3.46 \pm 0.02$ | $6.98 \pm 0.19$ |
| | DropChannel | $75.3 \pm 0.2$ | $1.02 \pm 0.01$ | $\mathbf{3.43 \pm 0.03}$ | $\mathbf{5.57 \pm 0.08}$ |
| | DropLayer | $\mathbf{75.8 \pm 0.3}$ | $1.04 \pm 0.02$ | $3.46 \pm 0.04$ | $7.42 \pm 0.32$ |
| | Omnibus dropout | $\mathbf{76.3 \pm 0.1}$ | $\mathbf{1.00 \pm 0.01}$ | $\mathbf{3.37 \pm 0.02}$ | $7.11 \pm 0.20$ |

Table 3: Results showing accuracy and uncertainty estimates produced by different types of **explicit ensembles** when $drop\_rate = 0.1$ for all models. five models are used for each ensemble. We generate 6 sampled models from each dropout models during evaluation (30 samples in total). The top-2 performing results for each metric is bold-faced.

| Datasets | Methods | Accuracy ↑ | NLL ↓ | Brier ↓ ($\times 10^{-3}$) | ECE ↓ ($\times 10^{-2}$) |
|---|---|---|---|---|---|
| SVHN | Deep Ensemble | $96.6 \pm 0.1$ | $0.179 \pm 0.009$ | $5.39 \pm 0.16$ | $1.08 \pm 0.08$ |
| | Dropout | $97.0 \pm 0.1$ | $0.141 \pm 0.006$ | $4.82 \pm 0.15$ | $\mathbf{0.736 \pm 0.077}$ |
| | DropBlock | $\mathbf{97.2 \pm 0.1}$ | $0.125 \pm 0.004$ | $4.79 \pm 0.13$ | $1.86 \pm 0.09$ |
| | DropChannel | $97.0 \pm 0.1$ | $0.129 \pm 0.004$ | $4.82 \pm 0.14$ | $0.949 \pm 0.082$ |
| | DropLayer | $96.8 \pm 0.1$ | $0.132 \pm 0.005$ | $4.91 \pm 0.14$ | $\mathbf{0.575 \pm 0.077}$ |
| | Omnibus dropout | $\mathbf{97.2 \pm 0.1}$ | $\mathbf{0.122 \pm 0.004}$ | $\mathbf{4.61 \pm 0.13}$ | $1.05 \pm 0.07$ |
| CIFAR10 | Deep Ensemble | $\mathbf{95.2 \pm 0.2}$ | $0.181 \pm 0.009$ | $\mathbf{7.40 \pm 0.28}$ | $1.40 \pm 0.16$ |
| | Dropout | $94.4 \pm 0.2$ | $0.176 \pm 0.008$ | $8.17 \pm 0.29$ | $1.04 \pm 0.16$ |
| | DropBlock | $94.0 \pm 0.2$ | $0.185 \pm 0.006$ | $9.12 \pm 0.28$ | $1.51 \pm 0.18$ |
| | DropChannel | $94.3 \pm 0.2$ | $0.174 \pm 0.007$ | $8.35 \pm 0.29$ | $\mathbf{0.900 \pm 0.152}$ |
| | DropLayer | $94.8 \pm 0.2$ | $\mathbf{0.173 \pm 0.006}$ | $7.88 \pm 0.26$ | $2.04 \pm 0.17$ |
| | Omnibus dropout | $\mathbf{94.8 \pm 0.2}$ | $\mathbf{0.160 \pm 0.006}$ | $\mathbf{7.82 \pm 0.27}$ | $0.953 \pm 0.156$ |
| CIFAR100 | Deep Ensemble | $78.0 \pm 0.4$ | $0.923 \pm 0.020$ | $3.12 \pm 0.05$ | $5.12 \pm 0.34$ |
| | Dropout | $77.5 \pm 0.4$ | $0.931 \pm 0.020$ | $3.18 \pm 0.05$ | $4.51 \pm 0.32$ |
| | DropBlock | $77.3 \pm 0.4$ | $0.909 \pm 0.019$ | $3.17 \pm 0.06$ | $3.23 \pm 0.31$ |
| | DropChannel | $77.1 \pm 0.4$ | $0.871 \pm 0.017$ | $3.13 \pm 0.04$ | $\mathbf{2.24 \pm 0.28}$ |
| | DropLayer | $\mathbf{78.1 \pm 0.4}$ | $\mathbf{0.855 \pm 0.017}$ | $\mathbf{3.05 \pm 0.04}$ | $\mathbf{2.14 \pm 0.28}$ |
| | Omnibus dropout | $\mathbf{78.0 \pm 0.4}$ | $0.863 \pm 0.017$ | $\mathbf{3.08 \pm 0.05}$ | $3.27 \pm 0.29$ |

As discussed in Section 4.1, patterns similar to the plots obtained on the CIFAR-10 dataset in Figure 3 are also observed consistently here. The only exception is to the *MC dropLayer* model on the SVHN dataset, which obtains better performance on individual model but much smaller improvements in both Brier score and test accuracy compared to the other the other dropout methods. We would like to highlight out that the seemingly contradictory results is likely caused by the shallow network used, an 18-layer ResNet. As no down-sampling layers are dropped out for *layer dropout*, the effective number of ResNet blocks that can be dropped is very small, leading to a much smaller dropout rate compared to ther other methods. This is not an issue with deeper models in which the number of downsampling blocks are much more than that of non-downsampling ones.

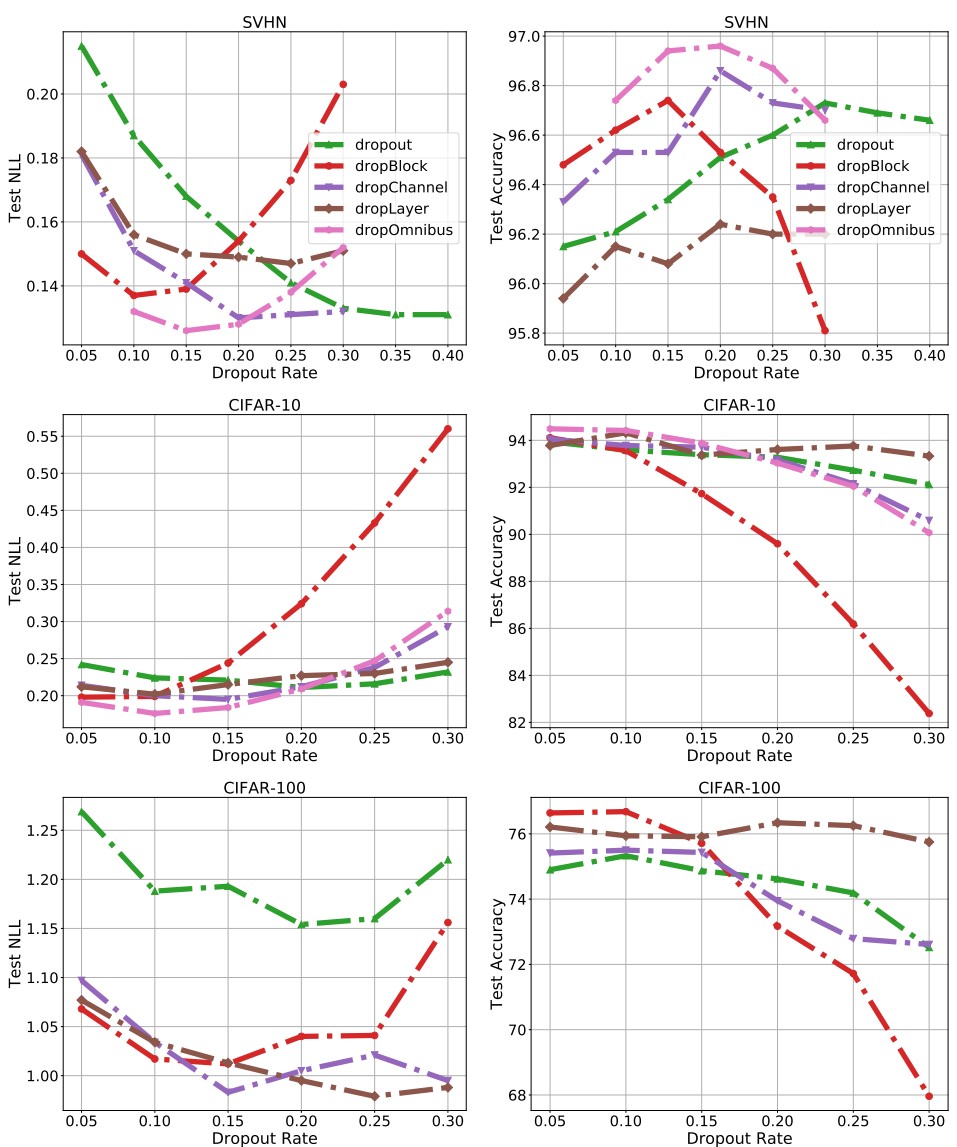

Figure 7: Plots of test time NLL (left) and accuracy (right) against dropout rate for models trained with different types of dropout on the SVHN, CIFAR-10 and CIFAR-100 datasets.

**Additional Results Using Explicit Dropout Ensembles.** Smith & Gal (2018) demonstrated that explicit ensembles with *MC dropout* models produce better uncertainty estimates than that of *deep ensembles*. Thus we examine the effectiveness of ensembling multiple explicit models trained with structured dropout. This would serve as a fair comparison to *deep ensemble*. We use five explicit models, each trained with random initialization. $drop\_rate = 0.1$ is used for all methods as it is impractical to tune for the optimal dropout rates for an ensemble of five models. Note that the optimal

dropout rates found for individual models do not carry over to explicit ensembles of five models, as we observe in our experiments. At test time, we generate six sampled models from each dropout models (30 samples in total).

Results obtained are summarized in Table 3. Similar to previous findings, ensembles with *omnibus dropout* consistently outperform all the rest. Moreover, ensembles with all types of structured dropout methods do better than *MC dropout* and *deep ensembles*. *Deep ensembles* does the worst in terms of uncertainty estimates.

We also investigate the sensitivity of the methods to the choice of dropout rate. To that end, we also report the results obtained with a single model for each method, with a fixed dropout rate of $0.1$, a reasonable default value for dropout rate in general. The results are shown in Table 2. Possibly due to the combination of all dropout method, *omnibus dropout* seems to be also relatively insensitive to the choice of dropout rate, performing well in all the experiments.

**Results on Tuning the Dropout Rate.** Figure 7 illustrates the plots of NLL and accuracy against dropout rate for all models on all of the datasets. As discussed in Appendix B, conflict between NLL and accuracy can occur sometimes. Interestingly, the NLL drastically increases after minima on all three datasets for *dropBlock*, suggesting the possibility that the block size for *dropBlock* may be too large towards later convolutional layers when the size of feature maps are comparable to that of block size.

