# OpenReview forum: "Omnibus Dropout for Improving The Probabilistic Classification Outputs of ConvNets"
_ICLR.cc/2020/Conference — Reject_

### Official Review · AnonReviewer1 · 2019-10-24
**Official Blind Review #1**

**Rating:** 3

**Review:**

This paper proposed to use multiple structured dropout techniques to improve the ensemble performance of convolutional neural networks as well as the calibration of their probabilistic predictions.

The approach, which is termed omnibus dropout and combines multiple existing dropout methods, is reasonable and straightforward.

The paper presents extensive experimental results and analyses by mainly comparing with the explicit ensemble of multiple neural networks. The experiments reveal interesting properties of the learned networks and compelling results.

My main concern is about the technical novelty of the paper. It does not supply a new method in essence and instead provides careful experimental studies, from both accuracy and calibration perspectives, about a combination of existing dropout techniques. It reads like a very solid workshop paper in my opinion, but it is probably not a good fit to the main conference.

Questions:
1. The reasoning in the first two paragraphs of the introduction is confusing or misleading. The first paragraph is mainly about the poorly calibrated probabilistic outputs of neural networks, while the second paragraph suddenly shifts to the performance of ensembled networks in terms of accuracy.

2. Some of the comparisons with the "deep ensemble" may be unfair. There are only five networks in "deep ensemble", but 30 are used in test time for the proposed method.

3. Where is "deep ensemble" in the active learning experiments? I could not find it in Figure 5.

**Experience Assessment:**

I have published in this field for several years.

**Review Assessment: Checking Correctness Of Derivations And Theory:**

I assessed the sensibility of the derivations and theory.

**Review Assessment: Checking Correctness Of Experiments:**

I carefully checked the experiments.

**Review Assessment: Thoroughness In Paper Reading:**

I read the paper thoroughly.

---

> ### Author Response · Authors · 2019-11-14
> **Response to Reviewer #1**
>
> We would like to thank you for taking precious time to review our paper.
>
> Firstly, we would like to address your main concern regarding the technical novelty of the paper. While we do agree that the proposed method of omnibus dropout is simple, we believe that the paper provides several significant contributions to the field of uncertainty estimation for deep learning as listed below.
> 1. While it was previously discovered empirically that deep ensembles lead to better uncertainty estimates compared to MC dropout, no justification had been given to explain the surprising observation that, even a simple frequentist approach of ensembling 5 neural networks can outperform some of the Bayesian approaches like MC dropout. In this paper, we attribute the lackluster performance of MC dropout to the lack of diversity in the predictions of models sampled through dropout. This is a novel perspective.
> 2. Although structured dropout techniques are widely used as regularization techniques to improve the accuracy of neural networks, they are not adopted for uncertainty estimation at all. In this paper, together with the theoretical justification based on ensemble diversity, we demonstrate the merit of using structured dropout methods at test time to obtain better uncertainty estimations as well.
> 3. Empirically we found that different structured dropout techniques have variable performance for different datasets and architectures, and searching for the best one can be expensive. We provide a simple solution which combines different kinds of dropout. We empirically demonstrate that this omnibus strategy yields state-of-the-art performance compared to widely used baselines for uncertainty estimates like temperature scaling, deep ensemble, and regular MC dropout.
>
> Response to specific questions:
> 1. We will modify the text to make the first two paragraphs flow more naturally. To clarify, we mention “ensembled networks” in the second paragraph as it was previously demonstrated that ensembling improves calibration performance as well, which is the main focus of our paper.
> 2. While we do agree that comparing an explicit ensemble of 5 models with 30 sampled models through dropout at test time may be unfair, this has been a widely adopted approach for comparison in the literature. Moreover, despite the fact that the proposed method incurs 6 times more time complexity during test time, we would like to highlight that our proposed method (and the baseline of MC dropout) is 5 times cheaper during training  and also 5 times cheaper in terms of memory demand as only one model needs to be saved and loaded instead of 5.
> 3. We did not incorporate experiments with deep ensembles in the active learning experiments. We will include it in the updated version of the paper.

---

### Official Review · AnonReviewer3 · 2019-10-26
**Official Blind Review #3**

**Rating:** 1

**Review:**

PAPER SUMMARY: The paper argues that (ensembles of models with different types of dropout applied to each model) perform better than (ensembles in which the same type of dropout is applied to each model). They attribute this to increasing model diversity in the former case, and experimentally validate their claims.

MAJOR COMMENTS:
1. Motivation Unclear:
- Notational issues in (2): MSE(h_t | x) involves y, but y is not specified in the definition. As a result, later E_{x}[…] is evaluated disregarding the dependence on y, which is ambiguous.
- Derivation of the second equality in (2) is not obvious, and needs a detailed proof. Notational issues exist in switching between H and h.
- The above is used to argue that “the more diverse the models, the better performance achieved”. It is unclear how this follows from (2).

2. Unclear / Imprecise Writing
- Before Sec. 3.4:  “This is because higher Dropout rates lead to smaller effective network capacities” Needs reference.
- “Dropout uncertainty can be obtained sequentially”. Unclear what this means.

3. The entire proposed method is described in one line, “we propose a novel omnibus dropout strategy, which merely combines all the aforementioned methods”. It is very unclear as to what the authors mean by combination.

4. Experiments
- It seems that Omnibus dropout leads to a moderate “diversity”, and improves ensemble performance only for SVHN, out of the three datasets tested in Figure 4. For SVHN, the improvement is 0.1% accuracy, which is within the standard error (0.1). Hence, it seems that the proposed method provides no performance improvement.
- A Similar statement is true for the other metrics (NLL, Brier Score, ECE).

Score [Scale of 1-10]:
3: Reject
The paper needs either more convincing experiments demonstrating the claims or theoretical analysis explaining the behavior for small networks. The paper needs to be rewritten to improve presentation, and state motivation, problem statement, and contributions clearly.


**Experience Assessment:**

I have published in this field for several years.

**Review Assessment: Checking Correctness Of Derivations And Theory:**

N/A

**Review Assessment: Checking Correctness Of Experiments:**

I assessed the sensibility of the experiments.

**Review Assessment: Thoroughness In Paper Reading:**

I read the paper thoroughly.

---

> ### Author Response · Authors · 2019-11-14
> **Response to reviewer #3**
>
> We would like to thank you for taking precious time to review our paper.
>
> “Motivation Unclear”
> 1.	We will fix the notational issue regarding MSE not involving y.
> 2.	We will incorporate a detailed proof of equation (2) in the appendix.
> 3.	From equation (2), we can see that MSE of the overall ensemble model H can be decomposed into the difference between average MSE of individual model h_t and the model ambiguity (a notion to measure diversity among predictions, as defined in the first paragraph of section 3.2) of the individual models. Hence, the more diverse the individual predictions, the larger reduction in terms of MSE, and hence the better calibrated the predictions are (MSE can be seen as a measure of both accuracy and calibration, as described in Appendix B of our paper). This motivates us to use structured dropout methods to enhance model diversity among the sampled model, thereby producing better-calibrated predictions.
>
> “Unclear / Imprecise Writing”
> 1.	Higher dropout rates lead to smaller effective network capacities because high dropout rates would lead to less amount of parameters in the neural networks on average, thereby reducing the model capacity. We will make this more explicit in the updated version of the paper.
> 2.	By “dropout uncertainty can be obtained sequentially”, we mean that we can iteratively get many samples by deploying dropout at test time to obtain uncertainty estimates. As a direct comparison to this, when deploying deep ensembles, multiple models have to be saved and loaded to obtain uncertainty estimates. We will make this more clear in the updated version of the paper as well.
> 3.	We describe in the following sentence that “The implementation of omnibus dropout involves the sequential execution of the nested group of dropout methods: regular dropLayer, dropChannel, dropBlock and regular dropout.” We will incorporate a figure to illustrate the architecture as well in the updated version of the paper.
>
> “Experiments”
> 1.	As we analyzed in the second paragraph of Section 4.1 of our paper, we empirically demonstrated that the moderate improvement in diversity is key to why omnibus dropout is consistently one of the best approaches. In terms of performance improvement, we would like to emphasize that, to the best of our knowledge, no prior work has considered any form of structured dropout for uncertainty estimation and the proposed technique can perform better than three of the most widely used strategies (MC dropout, deep ensemble, and temperature scaling) in terms of NLL, ECE and Brier score, as demonstrated in our experiments.

---

### Official Review · AnonReviewer4 · 2019-11-06
**Official Blind Review #4**

**Rating:** 1

**Review:**

Summary of the paper:
The paper tackles the probabilistic classification problem which is interesting and important in deep learning. The paper proposes an approach called omnibus dropout. Omnibus dropout is a sequential execution of existing dropout techniques such as drop layer, drop channel, drop block and regular dropout. The experimental results suggest that omnibus dropout perform reasonably well in various datasets.

Pros:
1. The proposed approach is reasonable which I found no surprising it produces reasonably well results.

Cons:
1. The paper could be improved in writing, especially on justification why this kind of dropout combination gives better performance.
2. The results are mixed and the improvements are not significant. So I am not convinced the proposed approach is always a better strategy.
3. The proposed approach is simply a combination of the existing dropout methods. The contribution of the paper is very limited.

Questions:
1. In table 1, different dropout rate is chosen for different method. I think it is also reasonable to compare different methods with the same dropout rate (basically this corresponds to the actual model capacity).

**Experience Assessment:**

I have read many papers in this area.

**Review Assessment: Checking Correctness Of Derivations And Theory:**

I assessed the sensibility of the derivations and theory.

**Review Assessment: Checking Correctness Of Experiments:**

I assessed the sensibility of the experiments.

**Review Assessment: Thoroughness In Paper Reading:**

I read the paper at least twice and used my best judgement in assessing the paper.

---

> ### Author Response · Authors · 2019-11-14
> **Response to Review #4**
>
> We would like to thank you for taking precious time to review our paper.
>
> 1.	We will make the justification as to why the omnibus dropout strategy improves the performance more clearly stated. In essence, we empirically find that structured dropout methods promote model diversity in the sampled models compared to regular dropout. This enhanced model diversity leads to more calibrated predictions, according to equation 2. However, as illustrated in Figure 3 of the paper, the improved model diversity provided by structured dropouts like dropBlock and dropChannel can come at the expense of reduced MSE of individual sampled models, thereby reducing the overall ensemble performance. An omnibus strategy strikes the right balance between the performance of the average individual sample performance and model diversity, as illustrated by low Brier score when n=1 and much larger improvement when the number of samples increases, thereby achieving better results overall.
>
> 2.	In terms of performance improvement, we would like to emphasize that, to the best of our knowledge, no prior work has considered any form of structured dropout for uncertainty estimation and the proposed technique can perform better than three of the most widely used strategies (MC dropout, deep ensemble, and temperature scaling) in terms of NLL, ECE and Brier score, as demonstrated in our experiments.
>
> 3.	We address your main concern regarding the technical novelty of the paper here. While we do agree that the proposed method of omnibus dropout is simple, we believe that the paper provides several significant contributions to the field of uncertainty estimation for deep learning as listed below.
> a.	While it was previously discovered empirically that deep ensembles lead to better uncertainty estimates compare to MC dropout, no justification has been given to explain this surprising observation that, even a simple frequentist approach of ensembling 5 neural networks can outperform some of the Bayesian approaches like MC dropout. In this paper, we attribute the lackluster performance of MC dropout to the lack of diversity in the predictions of models sampled through dropout.
> b.	Although structured dropout techniques are widely used as regularization techniques to improve the accuracy of neural networks, they are not adopted for uncertainty estimation at all. In this paper, together with the theoretical justification based on ensemble diversity, we demonstrate the merit of using structured dropout methods over the regular dropout method for uncertainty estimation as well.
> c.	Empirically we found that different structured dropout techniques have variable performance for different datasets and architectures, and searching for the best one can be expensive. We provide a simple solution which combines different kinds of dropout. We empirically demonstrate that this omnibus strategy yields state-of-the-art performance compared to widely used baselines for uncertainty estimates like temperature scaling, deep ensemble, and regular MC dropout.
>
> 4.	We also considered using a constant dropout rate of 0.1 for all methods, and the proposed method still performs consistently better than most of the strategies (see Appendix C Table 2)

---

### Decision · Program_Chairs · 2019-12-19

**Decision:**

Reject

**Comment:**

The paper investigates how to improve the performance of dropout and proposes an omnibus dropout strategy to reduce the correlation between the individual models.

All the reviewers felt that the paper requires more work before it can be accepted. In particular, the reviewers raised several concerns about novelty of the method relative to existing methods, significance of performance improvements and clarity of the presentation.

I encourage the authors to revise the draft based on the reviewers’ feedback and resubmit to a different venue.